# On the relation between transversal and longitudinal scaling in cities

**Fabiano L. Ribeiro**[1,2]*, **Joao Meirelles**[3]*, **Vinicius M. Netto**[4,5], **Camilo Rodrigues Neto**[6], **Andrea Baronchelli**[2,7]

**1** Department of Physics (DFI), Federal University of Lavras (UFLA), Lavras, MG, Brazil, **2** Department of Mathematics, City University of London, London, United Kingdom, **3** Department of Civil and Environmental Engineering, Swiss Federal Institute of Technology Lausanne, Lausanne, VD, Switzerland, **4** Department of Urbanism, Universidade Federal Fluminense (UFF), Niterói, RJ, Brasil, **5** Center for Urban Science and Progress, New York University (CUSP NYU), New York City, New York, United States of America, **6** School of Arts, Sciences and Humanities, University of Sao Paulo, Sao Paulo, SP, Brazil, **7** The Alan Turing Institute, British Library London, United kingdom

* fribeiro@ufla.br (FLR); joao.meirelles@epfl.ch (JM)

## Abstract

Does the scaling relationship between population sizes of cities with urban metrics like economic output and infrastructure (transversal scaling) mirror the evolution of individual cities in time (longitudinal scaling)? The answer to this question has important policy implications, but the lack of suitable data has so far hindered rigorous empirical tests. In this paper, we advance the debate by looking at the evolution of two urban variables, GDP and water network length, for over 5500 cities in Brazil. We find that longitudinal scaling exponents are city-specific. However, they are distributed around an average value that approaches the transversal scaling exponent provided that the data is decomposed to eliminate external factors, and only for cities with a sufficiently high growth rate. We also introduce a mathematical framework that connects the microscopic level to global behaviour, finding good agreement between theoretical predictions and empirical evidence in all analyzed cases. Our results add complexity to the idea that the longitudinal dynamics is a micro-scaling version of the transversal dynamics of the entire urban system. The longitudinal analysis can reveal differences in scaling behavior related to population size and nature of urban variables. Our approach also makes room for the role of external factors such as public policies and development, and opens up new possibilities in the research of the effects of scaling and contextual factors.

**Data Availability Statement:** All relevant data are within the paper and its Supporting Information files.

**Funding:** This work was supported by École polytechnique fédérale de Lausanne (Joao

## Introduction

An unprecedented abundance of data has significantly advanced our understanding of urban phenomena over the past few years [1–4]. These advances were also enabled by the work of many theorists from different areas, such as physicists, urbanists and complex systems scientists, among others, who brought new insights and theories to the field, resulting in a significant step towards a new science of cities [5].

Meirelles), Conselho Nacional de Desenvolvimento Científico e Tecnológico (Fabiano Lemes Ribeiro, process number: 405921/2016-0), Coordenação de Aperfeiçoamento de Pessoal de Nível Superior (Fabiano Lemes Ribeiro, process number: 88881.119533/2016-01). The funders had no role in study design, data collection and analysis, decision to publish, or preparation of the manuscript.

**Competing interests:** The authors have declared that no competing interests exist.

A crucial finding concerns the scaling properties of urban systems. Empirical evidence has shown that an urban variable, $Y$, scales with the population size $N$ of a city, obeying a power law of the kind $Y \propto N^\beta$, where $\beta$ is the scaling exponent quantifying how the urban metric reacts to the population increase [6–12]. On the one hand, the data revealed that socioeconomic urban variables such as the number of patents, wages, and GDP present a *superlinear* behavior in relation to the population size ($\beta > 1$). Using the language of economics, one might say that this kind of urban variables exhibits *increasing returns* to urban scale. On the other hand, infrastructure variables such as the length of streets and the number of gas stations scale *sublinearly* with the population size ($\beta < 1$). Finally, there is a third class of variables related to individual basic services, such as household electrical and water consumption, and total employment, which scales linearly with population size ($\beta \approx 1$).

Among the various attempts to explain such behavior in urban phenomena [13–15], one of the most successful was proposed by Bettencourt and colleagues [16]. Their theory proposes that urban scaling is a result of an interplay between urban density and diversity, which are related to economic competition and knowledge exchange, respectively. The value of a socioeconomic quantity would be a direct consequence of the number of human interactions in a city, which grows superlinearly with the population size [17]. On the other hand, infrastructure variables present scaling economies with the population size, in the sense that bigger cities need less infrastructure per capita. It results in a sublinear behaviour between infrastructure and population size. This urban scaling economy is analogous (but only qualitatively) to what happens in biological systems: larger animals are more economical energetically since they spend less energy per unit of mass [18].

Using large datasets, recent statistical findings on the superlinear behavior of cities seem to confirm earlier propositions and findings in spatial economics on agglomeration economies. A large body of work has empirically demonstrated the role of urban agglomeration in increasing returns to scale in different regional contexts [20–23, 54]. Efficiency increases when organizations progress from small-scale to large-scale production. Increasing returns provide an incentive to concentrate production, as transport costs are minimized by locations close to markets, and have an obvious centrality to urban patterns [19]. In addition to Marshall-Arrow-Romer scale externalities related to size, density and specialization [24], evidences of Jacobs' externalities have been found, as spatial effects of diversification on knowledge spillovers across industries concentrated in cities [23, 25, 26]. To sum up, the interaction between firms and the individuals that compose a city results in innovation, economic growth, and returns to scale.

As these scaling laws have been observed in different countries [6–8, 10, 27–32] and periods of time [33, 34], some works also claimed that such patterns are, in fact, the manifestation of a universal law that would generally govern cities regardless of their context, culture, geography, level of technology, policies or history [9, 16, 17, 30–32]. According to this proposition, in the long term, the general performance of a particular city would be largely independent of individual political choices. However, the universality proposition has been challenged, either by counter-evidence [10, 28, 35–37], or by methodological arguments [38–42]. Thus, the universality proposition is still an open question both from the methodological and the theoretical point of views. Addressing this issue is important in urban science also because the validation of such universal dynamics could help urban policymakers to identify opportunities to improve urban metrics. For instance, Alves et al. [43] have proposed a new index based on scaling laws instead of per-capita analysis to classify levels of economic efficiency of cities; Youn et al. [44] identify the relative abundance of business and economic differentiation according to city size; and Ribeiro et al. [17] discuss how improvements in mobility,

connections mobility between people, and scaling properties can contribute to production processes in cities (see also [9, 45, 46]).

A key open question is the difference between the scaling properties of *single* cities and *sets* of cities. In short, does an individual city growing in time follow the same scaling pattern observed for a snapshot of a group of cities? In the last years, few works have accurately focused on the dynamics of individual cities [47–51], while a growing literature has been concentrating on the scaling properties of sets of cities. We call the former *longitudinal* scaling properties, which take into account the evolution of individual cities in time, and the latter *transversal* scaling across an urban system, i.e., computed from the set of cities that compose the system. Some recent works addressed this issue, reaching no unanimous conclusion. For example, Depersin and Barthelemy analyzed the scaling exponent in time for delays in traffic congestion in 101 US cities, and found longitudinal scaling to be path-dependent on the individual evolution of cities and unrelated to the transversal scaling, challenging the universality proposition [49]. In turn, Hong et al. argued that longitudinal and transversal exponents are correlated, but it is essential to eliminate global effects to properly measure the longitudinal scaling exponent [50]. Another work has found that the power-law scaling of 32 major cities in China could adequately be characterized for both transversal and longitudinal scaling [51]. More recent work also analyzed the issue for the wage income in Sweden and found super-linear scaling for both longitudinal and transversal scaling, but the former was characterized by larger scaling exponents [48]. Most recently, Bettencourt et al. [47] proposed a mathematical analysis that shows explicitly how to connect the transversal and longitudinal exponent.

Here, we will present our analysis of the transversal and longitudinal behavior of GDP and water network length (socio-economic and infrastructure variables, respectively) for more than 5500 Brazilian cities. Our main results show that the longitudinal scaling exponents are different from each other, as suggested by Depersin and Barthelemy's work [49], but they are distributed around an average that approaches the transversal scaling exponent when the data is decomposed to eliminate external factors and when we consider only subsets of cities with a sufficiently large growth rate. Such results support the idea that the longitudinal dynamics is a micro-scaling version of the transversal dynamics of the entire urban system.

The paper is organized as follows: having posed the research problem in this section, we shall unfold our method and data used to assess the evolution of two different urban metrics in section *Materials and methods*. Section *Empirical Evidence* brings results from our data analysis for both transversal and longitudinal scaling for our studied variables, namely GDP and water network length as a function of population size in different periods of time (from 1998 to 2014) for all Brazilian cities. Section *Theoretical Approach* describes the dynamics of such properties as an analogous problem of particles in a vector field, applied in a way to render the relation between longitudinal and transversal scaling exponents clearer. It also explores the implications of our findings, along with potential contributions. Finally, we summarize our findings in section *Conclusions*.

## Materials and methods

### Data of the Brazilian urban system and its scaling properties

The available data (see spreadsheet in the supplementary material (SM)) cover all 5570 Brazilian municipalities. Data was collected from the website of the Brazilian Institute of Geography and Statistics (IBGE) Instituto Brasileiro de Geografia e Estatística, https://www.ibge.gov.br/ and from the water-sewage-waste companies national survey (SNIS) SNIS: National system of information on sanitation. Electronic version: app.cidades.gov.br/serieHistorica/. To be sure, 'municipality' is a political and territorial definition used for administrative purposes in Brazil,

and it includes both urban and rural areas within those borders. It does not necessarily represent *urban functional areas*, which eventually go beyond such borders, in conurbations. Therefore, we defined the urban unit of analysis as an aggregate set of municipalities that work as an integrated spatial and micro-economic agglomeration, i.e. sufficiently close to each other to form a single, continuous urban unit or formation. Each set aggregates the information of all municipalities that belong to it. By far, most of these sets are formed by a single municipality, but some of the biggest Brazilian urban agglomerations are formed by more than one municipality, aggregated in metropolitan conurbations. This definition has led us from 5570 administrative municipalities to 5507 urban agglomerations as 'aggregate sets of municipalities', which hereafter we will refer to simply as *cities*.

This approach will be restricted to two urban metrics, one for each scaling regime: (i) *GDP*, a socio-economic variable that typically presents a superlinear behavior with the population size, and (ii)*water supply network length*, an infrastructure variable which typically finds a sublinear behavior.

A recent work [10] has shown that over 60 variables for the Brazilian urban system are well described by a power-law equation of the form:

$$Y_i(t) = Y_0(t)N_i(t)^{\beta_T}. \tag{1}$$

Here, the time-dependent variables $Y_i(t)$ and $N_i(t)$ are relative to the city $i$; the former represents some urban metric (for instance GDP or water network length), and the latter represents the city population size. The two parameters in Eq (1) are the intercept parameter $Y_0(t)$ and the transversal scaling exponent $\beta_T$, which are obtained by the fit of this power law with the urban system data in a specific time $t$. These two parameters have to do with the *macro-scale* properties of the urban system and, at first, do not represent the particularities of a single city —the *micro-scale*. As we will show in the next sections, the intercept parameter is a time-dependent variable, while the transversal scaling exponent can or cannot be time-dependent.

## Results and discussion

### Empirical evidence

**Transversal scaling.** Fig 1 shows the GDP as a function of the population size for different years (from 1998 to 2014) in Brazilian cities. The straight lines in Fig 1 are the best fit, by the maximum likelihood method Here we consider that the probability $P[Y|N]$ for a specific city $i$ obeys a log-normal distribution with expected value $E[Y_i|N_i] = Y_0 N_i^{\beta}$. Then the log-likelihood for all cities, from the data set $\{(Y_i, N_i)\}_i$, is given by $\ln \mathcal{L} = \Sigma_i \ln P[Y_i|N_i] \propto \Sigma_i (\ln \alpha N_i^{\beta} - \ln Y_i)$, where the sum is over all the cities of the system. Then, we use Nelder-Mead method to find $\alpha$ and $\beta$ which maximize $\ln \mathcal{L}$. Some works in the recent literature suggest this methodology, as for instance [31, 52, 53]., of the Eq (1) for different years. The transversal scaling exponent $\beta_T$ (the slope) of each line in Fig 1 is always greater than one, indicating a persistent superlinear behavior. Moreover, the best fit lines are visually parallel, that is, $\beta_T$ is approximately constant, even with the time evolution of the cities, which reveals the robustness of the scaling exponent (see the video in the supplementary material SM2). These facts can be observed in more detail in Fig 2-a, which presents the time evolution of $\beta_T$, which stays approximately constant even with the intercept parameter $Y_0(t)$ continuously increasing with time (see Fig 2-b).

Fig 2 also presents the time evolution of the transversal scaling exponent $\beta_T$ for the water supply network length (in blue). In this case, $\beta_T$ is not constant and decreases over time, as seen in Fig 2-a while remaining smaller than 1, which is expected given it refers to an infrastructure variable. According to the available data, it is hard to establish whether its value will

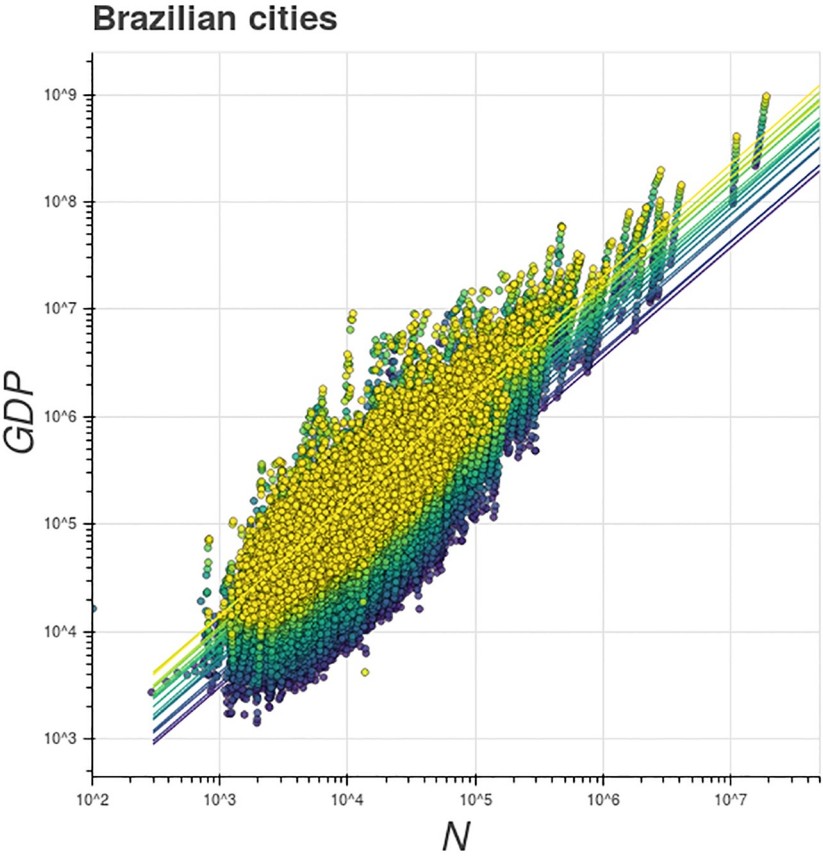

**Fig 1. Transversal scaling of GDP.** Scaling relation between population and GDP of Brazilian cities, from 1998 (blue) to 2014 (yellow). The straight lines are the best power-law equation fits for each year (by the maximum likelihood method). The straight lines are virtually parallel, which shows that the transversal scaling exponent is constant and robust. The scaling exponent is always greater than 1 for all years, with a mean $\bar{\beta}_T = 1.04$. It reveals a superlinear scaling property, compatible with the fact that the GDP is a socio-economic urban variable. The numeric time evolution of the transversal scaling exponent and the intercept parameter are shown in Fig 2.

stabilize or not. The fact that this variable is not constant could suggest that the urban system is still out of balance with respect to this urban metric, as suggested by Pumain et al.'s theory [14]. Moreover, the data suggest that the intercept parameter $Y_0(t)$ of this urban metric, as it was observed in GDP, maintains a continuous growth through the observed time frame (see Fig 2-c).

**Longitudinal scaling.** We now focus on the individual evolution of Brazilian cities. Fig 3 presents different ways of observing the longitudinal dynamics of the GDP and city population size. Fig 3-a presents the raw longitudinal trajectories, while Fig 3-b presents them re-scaled as $Y_i(t)/Y_i(t_0)$, as a function of $N_i(t)/N_i(t_0)$, following the idea proposed in [49]. The re-scaled form allows us to compare in one single image the slopes of the cities' trajectories. Here, $t_0$ is the first year that the data is available. One can see that cities experience different slopes, and in all cases, the exponent is greater than the transversal one (given by $\beta_T$ and represented by the dark red line in Fig 3-b). Similar evidence was reported recently by Depersin and Barthelemy [49], which analyzed the temporal dynamics of delay in traffic congestion in US cities. They observed that the individual dynamics do not collapse in a single and universal curve,

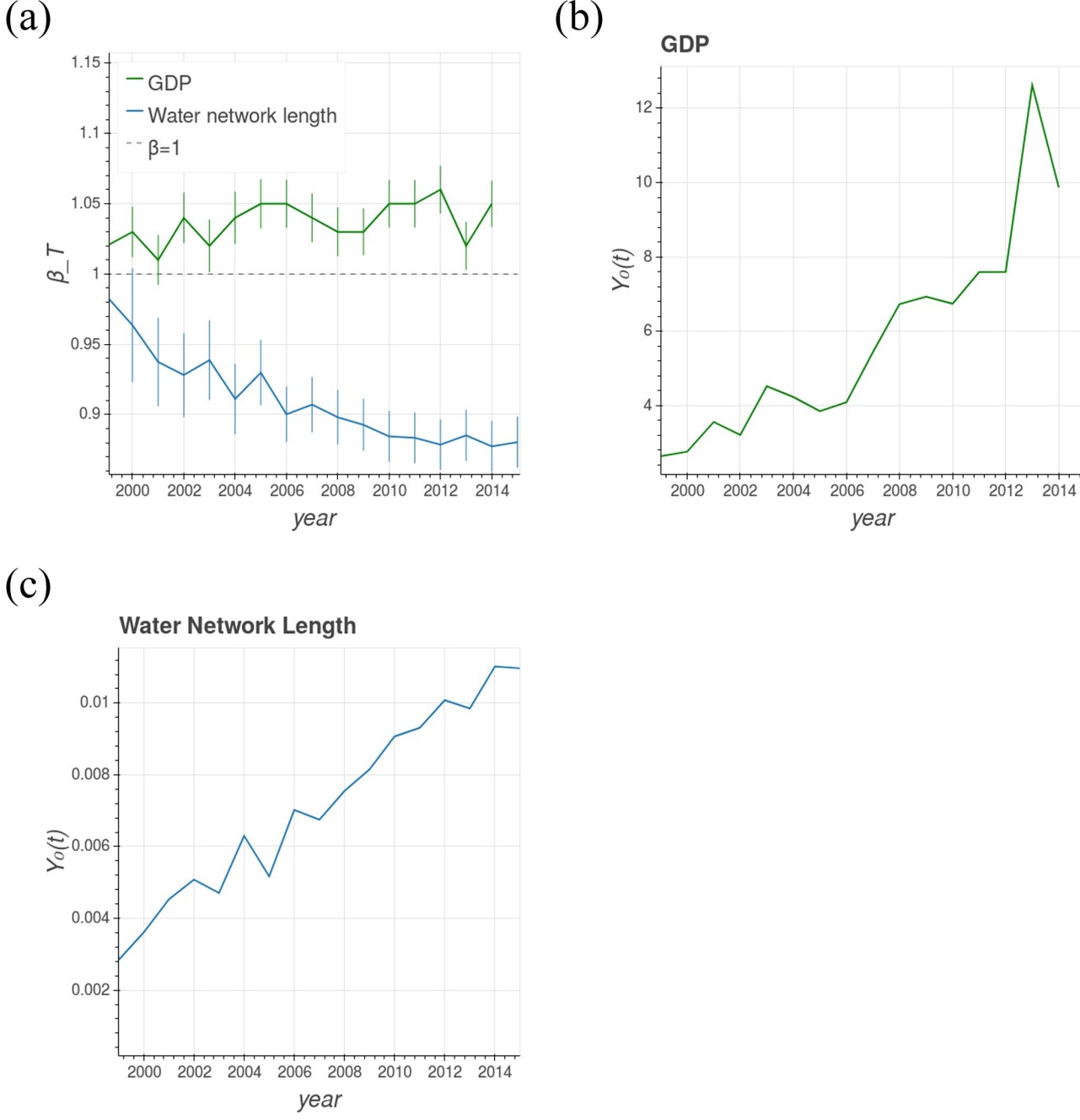

**Fig 2. Time evolution of transversal scaling.** a) Time evolution of the transversal scaling exponent $\beta_T$ for the GDP (green) and water supply network length (blue) for Brazilian cities. In the GDP case, there is no significant change of this parameter over the years and the regime (superlinear) is always sustained. In the case of water supply network length, $\beta_T$ is always smaller than 1, which is expected, given the infrastructure nature of this metric, and it is decreasing over time. b) and c) present the time evolution of the intercept parameter $Y_0(t)$ for GDP and water supply network length, respectively. The intercept parameter is constantly growing for both urban metrics.

and suggested that longitudinal scaling in cities is not governed by a single universal scaling exponent as the global system is. However, the data that we are analysing here suggest the longitudinal exponents are distributed around an average value which is compatible with the transversal scaling exponent if some restrictions are satisfied.

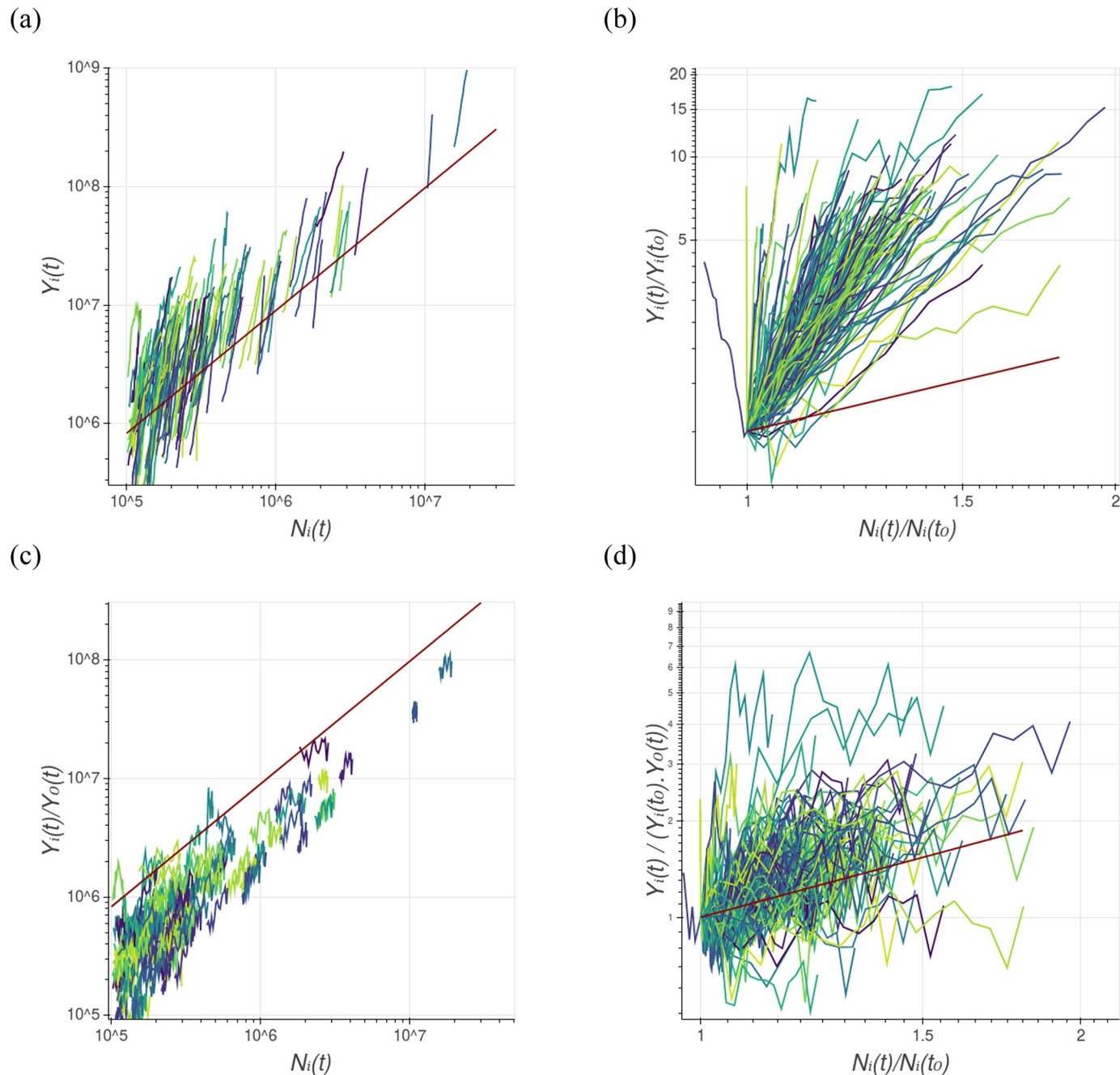

**Fig 3. Normalization of longitudinal scaling.** Different ways to see the longitudinal dynamics of GDP and population size for all Brazilian cities. Each trajectory represents the time evolution of one single urban area, from the year 1998 to 2014. a) log-log plot of the time evolution of the raw data of GDP as a function of population size. The dark red straight line is the power-law equation, with the average transversal scaling exponent $\bar{\beta}_T = 1.04$. b) log-log plot of the re-scaled form of the longitudinal dynamics, which allows us to compare the slopes of the cities' trajectory. This graph shows us that cities have different slopes, and they are greater than $\bar{\beta}_T$, represented by the dark red line. c) Decomposed longitudinal trajectory, which allows seeing the dynamics without global effects. d) Decomposed and re-scaled form of the longitudinal dynamics, which shows that the individual slopes are compatible with the transversal scaling exponent, represented by the dark red line. The distribution of the individual slopes (for raw and decomposed data) can be seen in Fig 4.

Individual cities are being pushed by the growth of the global intercept parameter $Y_i(t)$ and will rise in the $\ln Y$–$x$–$\ln N$ plane, having higher slopes than the global one. One way to deal with this is to *decompose* the longitudinal trajectory, graphing not $\ln Y_i(t)$ in the ordinate, but instead, $\ln Y_i(t) - \ln Y_0(t)$, that is $\ln(Y_i(t)/Y_0(t))$, in order to eliminate global effects, as suggested by [50]. The decomposed longitudinal trajectory is shown in Fig 3-c, and its re-scaled form is presented in Fig 3-d. The slopes observed on the decomposed and re-scaled form of the longitudinal trajectories are compatible with the transversal slope, represented by the dark red line in Fig (3-d).

Let us call $\beta_i$ the scaling exponent of the $i$-th city, that is, the slope of the (raw) trajectories described in Fig 3a and 3b calculated using the longitudinal evolution of $Y_i(t)$ with $N_i(t)$. Similarly, we can compute the individual decomposed scaling exponent, say $\beta_i^{dec}$, with the decomposed longitudinal trajectory described in Fig 3c and 3d. Fig 4 presents the distribution of the individual slope, for both sets $\{\beta_i\}_i$ and $\{\beta_i^{dec}\}_i$, for GDP and water supply network length, for all studied cities. One can see that the decomposed individual slopes for GDP are distributed around the global slope, suggesting that the decomposed version of individual trajectories recover the transversal phenomena for GDP in Brazilian cities. Moreover, it suggests that regardless of each municipality having different dynamics, that is, different longitudinal scaling exponents $\beta_i^{dec}$, their distribution presents a mean value compatible with the transversal scaling exponent.

However, in the case of the water supply network length, the average of the distribution of the non-decomposed data is closer to the transversal slope than the decomposed one, suggesting that decomposition does not recover the transversal scaling exponent for every urban variable. This may be the case since the transversal scaling exponent $\beta_T$ for water network length is

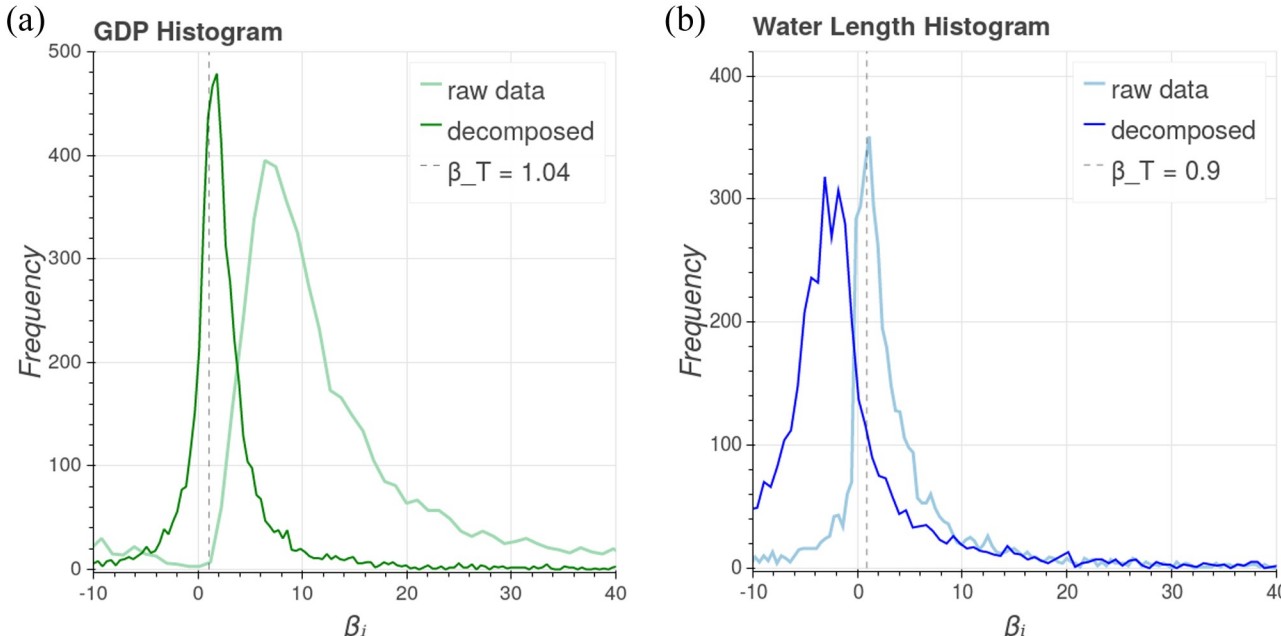

**Fig 4. Histogram of the longitudinal scaling exponent sets $\{\beta_i\}_i$ (raw data) and $\{\beta_i^{dec}\}_i$ (decomposed data) for GDP and water network length for the Brazilian cities.** For GDP (on the left), the decomposed data is distributed around the transversal scaling exponent $\beta_T$ (vertical dashed line), suggesting that it makes sense to decompose this urban variable. However, in the case of water network length (on the right), the distribution of the raw (non-decomposed) data is closer to the global slope than the distribution of the decomposed one, suggesting that decomposition is not working for this urban variable.

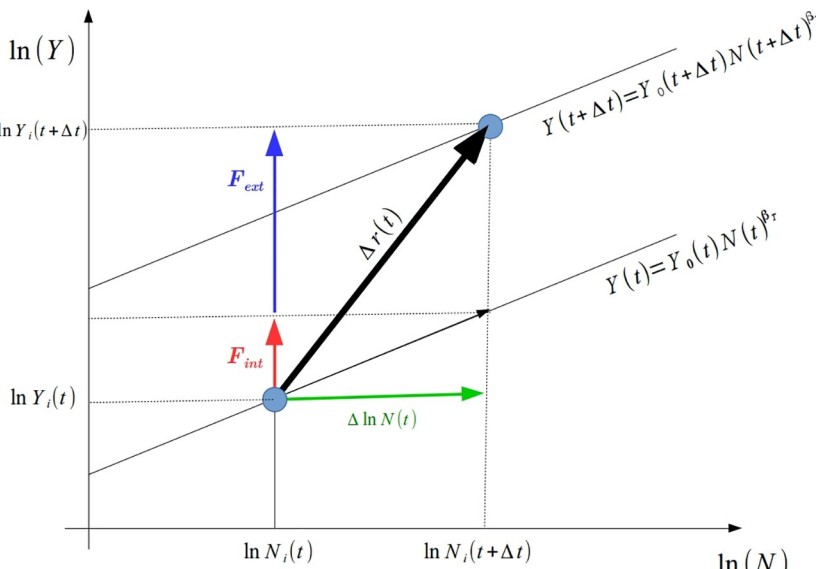

**Fig 5. Plane ln $Y$-x-ln $N$, representing the "movement" of the city as a particle in a vector field.** In the horizontal direction, we have the vector (green) that represents the increase in population size. In the vertical direction there is the action of two vectors: $F_{int}$ (red), which is an extensive quantity whose magnitude is a direct response to the increment of the population size related to an agglomeration effect between the individuals that live in this single city; and $F_{ext}$ (blue), which is the vector related to some external aspects, or the interaction between the individuals from this city with individuals of other cities, or some individual incorporated ability. The action of this vector field during a time interval $\Delta t$ conducts to a "displacement" $\Delta r(t)$ of this city (or particle) in this two-dimensional plane. The two parallel lines are given by the global system (transversal) power law (Eq (1)) in $t$ and $t + \Delta t$.

not stable across the studied years. These results suggest that the decomposition alone is not enough to infer that the individual and the global systems follow the same scaling properties in every case. In the next sections, we will introduce a theoretical approach that suggests that, in order to have an agreement between transversal and longitudinal scaling, it is necessary to consider a new ingredient: *the city growth rate.*

## Theoretical approach

In this section, we introduce a theoretical approach to describe the dynamics of urban metrics. In order to do so, we will treat the dynamics as an analogous problem of particles in a vector field. Fig 5 presents the plane ln $Y$-x-ln $N$ and the two-dimensional "movement" of one single city—a "particle"—as a result of the *vectors* acting in the horizontal or vertical direction.

In the horizontal direction there is a vector representing an increase in the city's population size. It is colored in green in Fig 5 and has a magnitude $\Delta \ln N_i(t)$, where we have introduced the compacted notation:

$$\Delta \ln N_i(t) \equiv \ln N_i(t + \Delta t) - \ln N_i(t), \tag{2}$$

as suggested in [50]. In the vertical direction of this plane, we have vectors acting on the increment of the urban metric (GDP or water supply network length, for instance). We will consider, by hypothesis, that there are at least two distinct vectors acting in this direction. The first, let's say $F_{int}$, represented by the red vector in Fig 5, is an *extensive* quantity whose magnitude is a direct response to the increase in population size. This vector has to do with the *agglomeration effect* stemming from the interaction between individuals or agencies located in

this single city. The second, let's say $F_{ext}$, represented by the blue vector in Fig 5, is the result of all external mechanisms such as, for instance, some wealth/knowledge that comes from other cities or regions; it can also represent the result of the interaction between individuals located in this single city with dwellers from other cities; or even some incorporated ability that increases individual productivity. This vectorial approach can be seen as a technical possibility to separate in two parts the forces acting in an urban metric: one that is pure scaling (internal factors); and the other that is the result of public policies or other external factors.

Therefore, the resulting vector acting on the vertical direction of the plane, say $F_{tot}$, is the sum of these two vectors, that is:

$$F_{tot} = F_{int} + F_{ext}, \tag{3}$$

which has a magnitude

$$F_{tot} = \Delta \ln Y_i(t) \equiv \ln Y_i(t + \Delta t) - \ln Y_i(t). \tag{4}$$

The action of these vector fields (in the horizontal and vertical directions) during a time interval $\Delta t$ conducts to a "displacement" $\Delta r(t)$ of this city (or particle) in the two-dimensional plane ln $Y$-x-ln $N$.

Now let us try to identify these vectors with the empirical variables available. The data presented in the previous section suggest that we have an empirical law that governs cities, which can be described by the expression (1). If this equation is a law, then any theory that is formulated to describe scaling properties in cities must be constrained to follow it. As this equation holds for any time $t$, we can write it for the next time instant $t + \Delta t$, that is:

$$Y_i(t + \Delta t) = Y_0(t + \Delta t)N_i(t + \Delta t)^{\beta_T(t+\Delta t)}. \tag{5}$$

Then, by extracting the logarithm of the ratio $Y_i(t + \Delta t)/Y_i(t)$ and using Eqs (1) and (5) we are conducted to:

$$\Delta \log Y_i(t) = \log\left(\frac{Y_0(t + \Delta t)}{Y_0(t)}\right) + (\bar{\beta}_T + \epsilon)\Delta \log N_i(t). \tag{6}$$

where we used the compacted forms defined on (2) and (4). Moreover, we also introduced $\bar{\beta}_T$ as the average value of the transversal exponent during the time interval $\Delta t$, and the parameter $\epsilon$, which is a quantity proportional to the difference $\beta_T(t + \Delta t) - \beta_T(t)$. In fact, the data analysis suggests that $\epsilon$ is sufficiently small for the cases we are studying here, so it will be neglected in our analyses. When $\beta_T(t)$ is constant, as it is approximately the case for GDP dynamics, then $\epsilon$ = 0.

The elements of Eq (6) can be identified with the vectors presented in Fig 5 and consequently with Eq (3). It allows us to identify:

$$F_{ext} = \log\left(\frac{Y_0(t + \Delta t)}{Y_0(t)}\right) \tag{7}$$

and

$$F_{int} = (\bar{\beta}_T + \epsilon)\Delta \log N(t). \tag{8}$$

The external vector, since it is directly computed from the ratio between the final and initial intercept parameter, can be interpreted as a measurement of the global growth of the urban metric. In this sense, the value given by (7) is an average value of the external vector. That is: typically, a city in the system has an external vector magnitude given by the value computed

from (7). In the previous section, when we decomposed each city's evolution into a relative change, we removed external factors acting on each city and considering only internal factors (the ones that come from agglomeration/scaling effects). However, we also saw that decomposition alone is not sufficient to establish a general relation between transversal and longitudinal scaling. In relation to the magnitude of the internal vector, it is an extensive variable; that is, it is a direct response to the increase of the population size. These results suggest that, in order for the urban metric to depend only on the population size (under the form $Y = cte \cdot N^{\beta_T}$), it is necessary for $\beta_T$ to be constant ($\epsilon \to 0$) and $F_{ext} \to 0$, which means absence of global growth. That can be the case for some urban metrics, but of course, it is not the case for GDP and many other variables. Our theoretical approach suggest that $\bar{\beta}_i^{dec} \neq \beta_T$ when $\epsilon \neq 0$, which was observed in our empirical data for the water supply network length.

**Relation between transversal and longitudinal scaling exponents.** With the approach presented above, it is possible to write a relation between the transversal and the longitudinal scaling exponent. Given that the longitudinal scaling exponent $\beta_i$ is obtained by:

$$\beta_i = \frac{\Delta \ln Y_i(t)}{\Delta \ln N_i(t)}, \tag{9}$$

then if we divide Eq (6) by $\Delta \ln N_i(t)$, we have:

$$\beta_i = \beta_T + \epsilon + \frac{F_{ext}}{\ln b_i}, \tag{10}$$

where $b_i \equiv N_i(t + \Delta t)/N_i(t)$ is the city population growth rate. The graphs in Fig 6 show that this result works very well when we analyze $\beta_i$ as a function of $b_i$, for both GDP and water

(a)

(b)

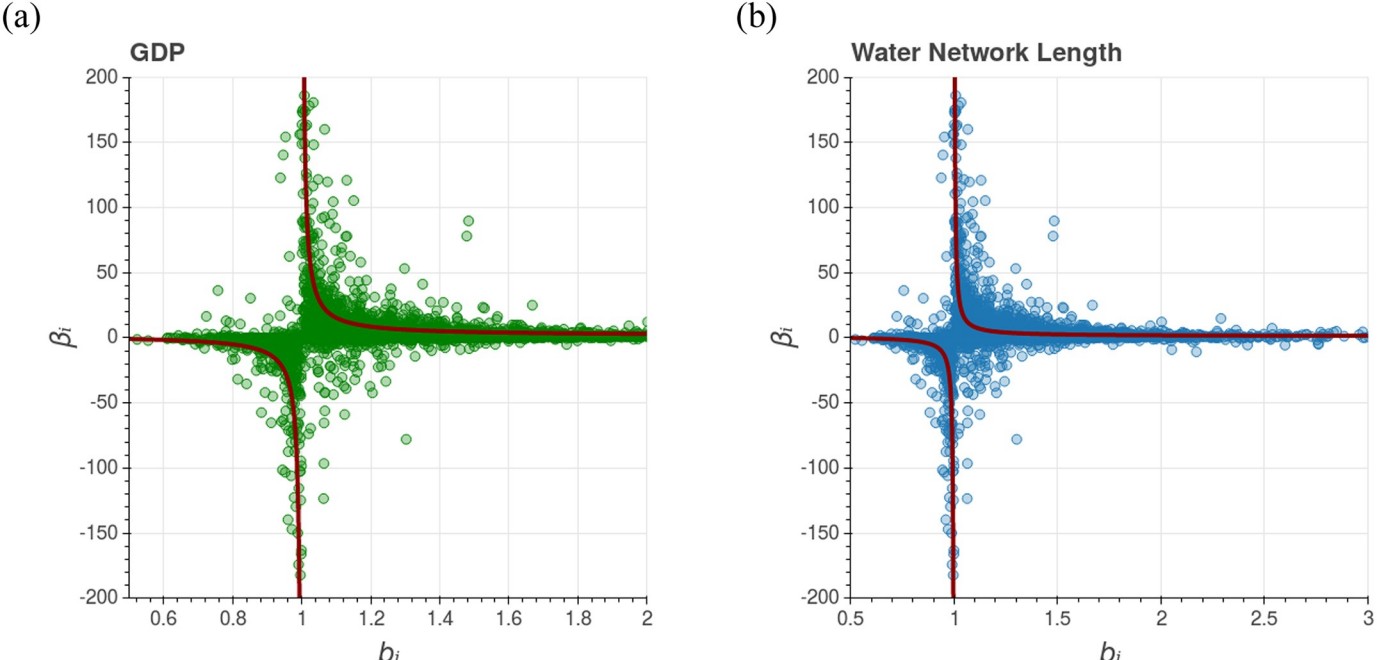

**Fig 6. Graph of $\beta_i$ as a function of the population growth rate $b_i$.** GDP (on the left) and water network length (on the right). Each dot represents the data of a single Brazilian city and the red curve is the theoretical prediction given by Eq (10) using: $\epsilon = 0$ for both cases; $\beta_T = 1.15$ for GDP; and $\beta_T = 0.9$ for water network length. This result illustrates the strong dependence between the longitudinal scaling exponent and the population growth rate of the city. It also suggests that cities with bigger $\beta_i$ are the ones with little or no growth ($b_i \approx 1$). Moreover, $b_i < 1$ (decreasing population) implies a negative $\beta_i$.

supply network length for the studied municipalities. It shows a strong dependence between these two variables. The Eq 10 also establishes a quantitative relation between transversal and longitudinal scaling, but one that only makes sense if $b_i \neq 1$. In short, it only works for cities that sufficiently grow during a time interval. It is important to note that cities that have a small growth rate ($b_i \approx 1$) will present a divergent or unstable longitudinal scaling exponent. Those cases behave differently from the pattern uncovered by our approach. This result (10) was also obtained by Bettencourt et al. [47] using a slightly different approach.

The result (10) also suggests that if $F_{ext} > 0$ and $b_i > 1$, which means that both the intercept parameter (global growth) and the population are growing in time, then $\beta_i$ will always be greater than the global exponent $\beta_T$. The increment in the intercept implies a more accentuated slope of the city trajectory in the plane ln $Y$-x-ln $N$ (that is, bigger $\beta_i$) in relation to the transversal trajectory (related to $\beta_T$), in accordance with empirical observations brought by our study as well as other evidences available in recent literature [49–51]. To sum up, the transversal and longitudinal scaling exponent will only be the same when:

1. $F_{ext} = 0$, that means only internal factors (agglomeration/scaling effects) are acting on the system;

2. $\epsilon = 0$, that means that $\beta_T$ is constant;

3. $b_i \neq 1$, i.e., population must to have a sufficient growth.

This conclusion is in agreement with [47].

Another interesting aspect is that, according to the plot presented in Fig 6, Eq (10) is compatible with the data even for diminishing cities ($b_i < 1$). That is, the relation between $\beta_i$ and $b_i$ also works for negative population growth. However, this result does not allow us to reach a definitive conclusion about the transversal and logitudinal scaling for diminishing cities. The vast majority of the Brazilian shrinking cities have small population sizes, Data shows that 70% of declining Brazilian cities have less than 10,000 inhabitants, and only 3 of them have more than 100,000 inhabitants. a growth rate around $b_i = 1$ and consequently an unstable longitudinal scaling exponent.

In urban scaling analysis, it is important to know the value of the scaling exponent. Since the exponent shows how the urban metric reacts to an increase in population, it gives us the efficiency and productivity of the city or the urban system (given by $\beta_i$ and $\beta_T$, respectively). For instance, in socio-economic variables, larger values of $\beta$ mean a more productive city, and for infrastructure variable, smaller values means a more efficient city. However, in the context we are analyzing, cities with very large values of $\beta_i$ are not necessarily more productive. In fact, large values of scaling exponents are related to cities with a very low growth rate (according to Eq (9)), so the raw value of $\beta_i$ is not useful if the purpose is to see how productivity or scaling economies emerge.

However, we believe that when the value of $\beta_i$ has a sufficiently large population growth rate, it will inform about the city's internal efficiency. In order to investigate this, we computed the average values of the longitudinal scaling exponents $\bar{\beta}_i$, using only cities with $b_i$ greater than a threshold $b_c$, and later built the graph presented in Fig 7, where we can see that $\bar{\beta}_i$ decreases drastically for greater values of $b_c$, approaching the transversal exponent value for both GDP and water network length. This result suggests that when we consider a city that has grown significantly during the time period analyzed, it is relevant to understand its longitudinal scaling growth properties as a microscopic version of the macroscopic growth of the urban system. Another important aspect of these findings is that only *decomposition* is not enough to link globally with longitudinal scaling, as highlighted in the last section. In fact, decomposition

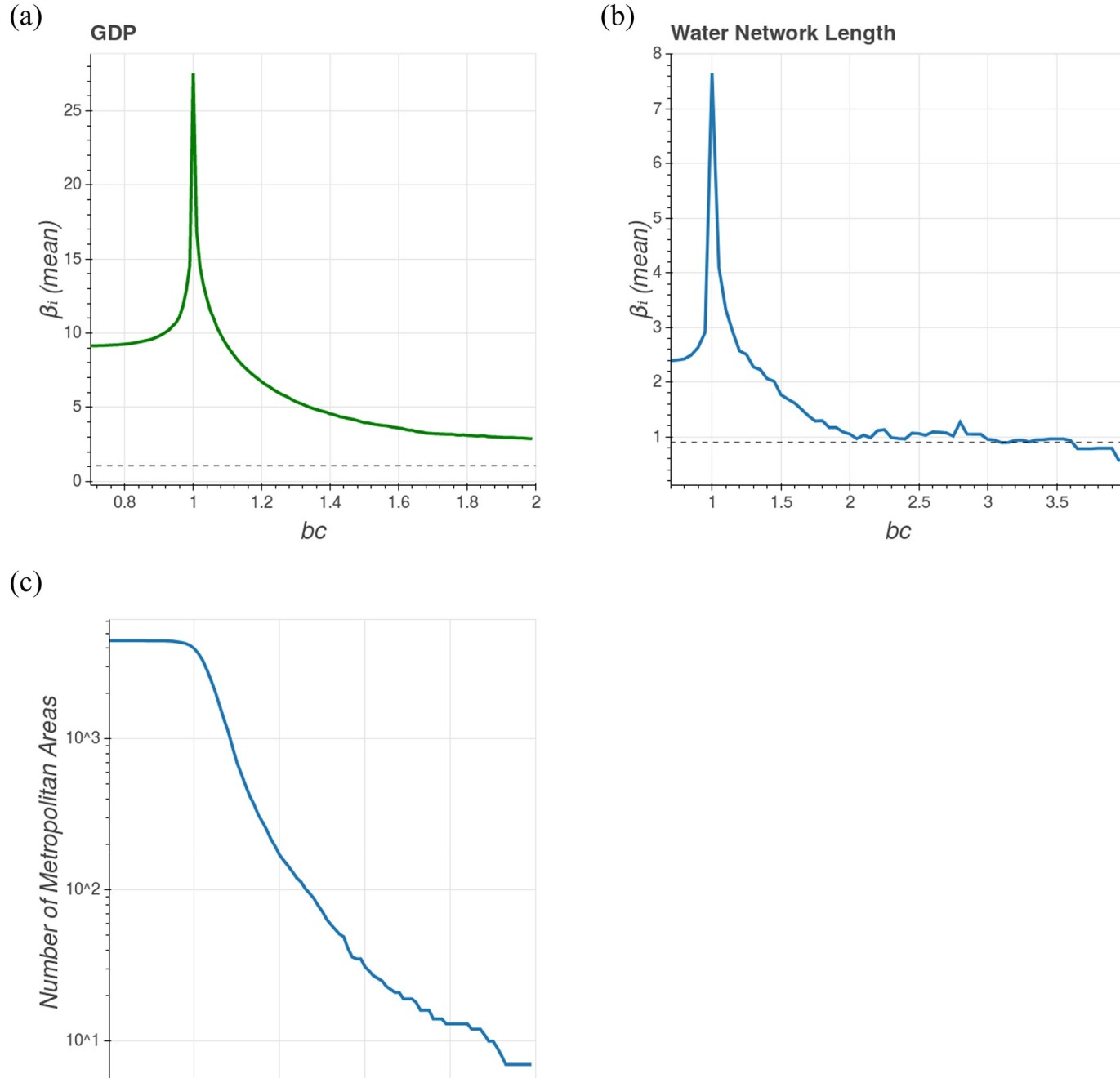

**Fig 7. Mean value of the longitudinal scaling exponent as a function of the growth rate threshold $b_c$.** For GDP (plot A) and water network length (plot B). The parameter $b_c$ delimits the cities that will be used to compute the average. The mean value of $\bar{\beta}_i$ decreases drastically as $b_c$ increases. Moreover the greater $b_c$ is the more $\bar{\beta}_i$ approaches to $\beta_T$ (represented by the dashed line). It's reasonable to think that in an ideal situation with no external forces and with a significant number of cities with the larger growth rate for better statistics, the average of the longitudinal scaling exponent will converge to the value of the transversal scaling exponent. Plot C: number of metropolitan areas used to compute the mean as a function of $b_c$. This number is drastically smaller for greater $b_c$ values.

only makes sense if we consider cities with sufficient growth in a given period of time, or with constant transversal scaling exponent $\beta_T$ ($\epsilon \neq 0$).

The results presented here must be confronted with more urban metrics and other countries. Moreover, a problem resulting from the approach presented in this section concerns the small number of municipalities that present $b_c$ sufficiently large. For instance, in order to compute the average $\bar{\beta}_i$ for $b_c = 4$ we used only 13 cities (see Fig 7-c). The statistics could be improved if we were studying an urban system with more cities experiencing higher growth rates, but maybe such systems don't even exist. Thus, a more feasible situation for future analyses consists of finding a way to normalize the longitudinal scaling exponent to the city's growth rate.

**The external vector.**   Eq (7) represents the average magnitude of the external vector, i.e. a city within the system will have an external vector with magnitude typically given by this value. However, it is interesting to discuss the specific external vector value acting on an individual city. This is a very difficult matter to be resolved given that it involves particularities of each city, but we can infer this answer from the data that we have available. For instance, we can use the result given by Eq (10) to infer the external vector $F^i_{ext}$ relative to the $i$-th city. That is, we can write that:

$$F^i_{ext} = (\beta_i - \beta_T - \epsilon) \ln b_i, \tag{11}$$

and if $\beta_T$, $\beta_i$ (given by Eq (9)) and $b_i$ are known, then it is possible to estimate (assuming $\epsilon \approx 0$) the individual external vector. That is the case presented by Fig 8, where each dot represents

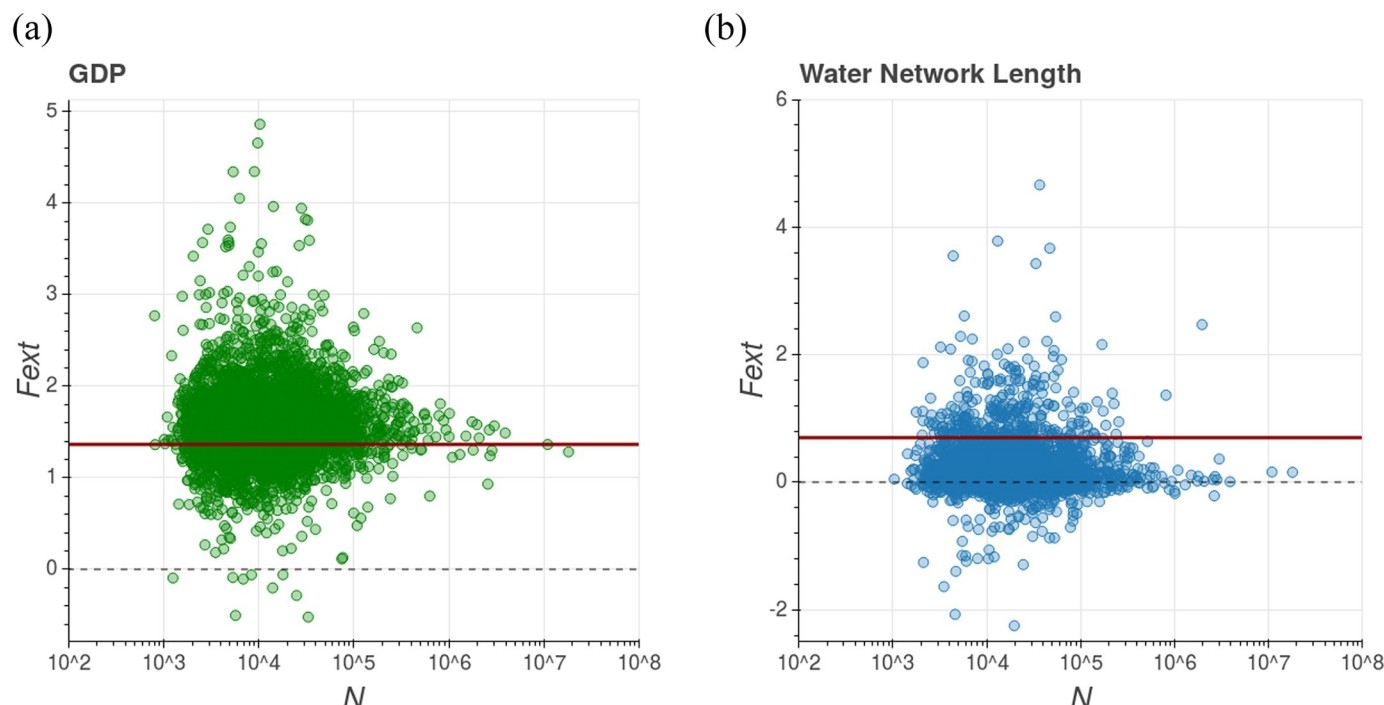

**Fig 8. Magnitude of the external vector as a function of city population size.** On the left, we have data referring to GDP and on the right to water network length. The dots represent $F^i_{ext}$ computed from the expression (11) while the red line represents the average magnitude of this vector over the system, computed by the expression (7). The dashed line is $F_{ext} = 0$. In the case of GDP, the municipality subsets of all sizes are distributed around the average value, but in the case of the water network length, the greater subsets presented external vector smaller than the average. These particularities imply different dynamics of the transversal scaling exponent, as shown in Fig 9.

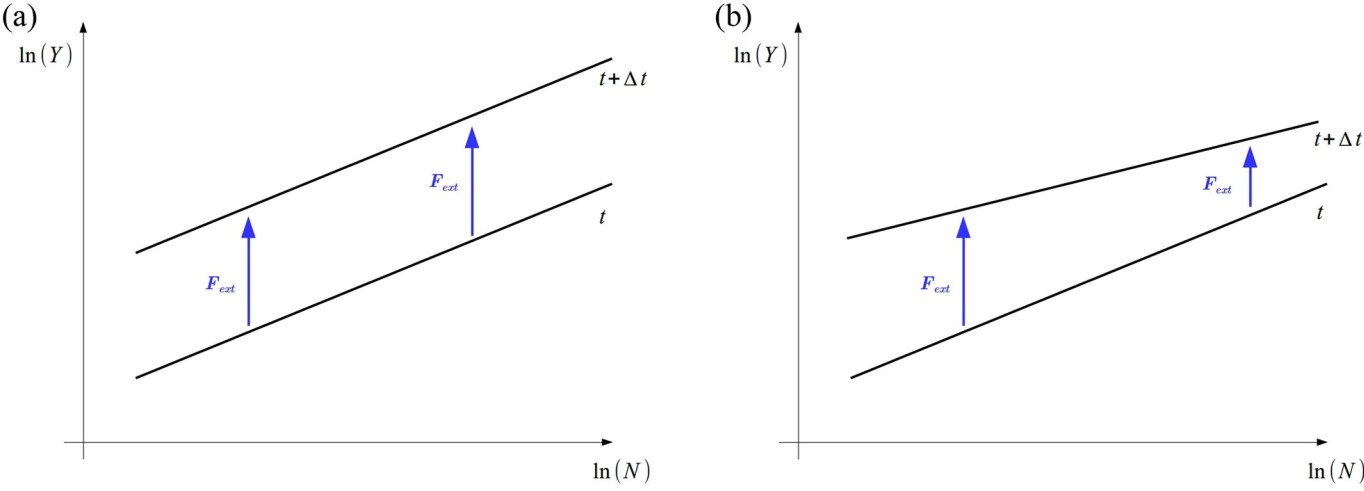

**Fig 9. Schematic drawing representing the plane ln $Y$-x-ln $N$ with two scenarios for the external vectors.** Magnitude of the external vector as a function of population size for every single Brazilian municipality in our subset. In the first scenario (on the right), the magnitude of the external vectors is the same regardless of city size; implying that the slope of the fit line (in ln $Y$-x-ln $N$ plane) remains constant from $t$ to $t + \Delta t$. That is more or less what happens in the GDP of the Brazilian municipalities, revealing that this urban variable is in a mature state in the system. In the second scenario (on the left), the external vector is smaller for bigger cities, which implies that the slope of the fit line decreases with time (from $t$ to $\Delta t$). That is apparently the case for the water network length of the Brazilian municipalities, suggesting that this urban metric is not mature in the system.

the value obtained for the external vector of a single city, for both GDP and water network length.

Fig 8 also brings the comparison between this individual and the average external vector magnitudes. It suggests an interesting aspect differentiating these two urban metrics' dynamics. In the case of GDP, cities of all sizes are distributed around the average magnitude of the external vector. However, in the case of water supply network length, bigger cities show external vectors smaller than the average. These characteristics imply different dynamics with respect to the transversal exponent, according to the schematic drawing in Fig 9, which presents the plane ln $Y$-x-ln $N$ with two scenarios for the external vectors. In the first scenario, the external vector is approximately the same for all cities of the system, regardless of their size; it implies that the slope of the fit line (in ln $Y$-x-ln $N$ plane) remains constant. That is more or less what happens in the GDP context of Brazilian cities. It suggests an equilibrium situation, or at least that this urban variable is in a mature state inside the system.

In the second scenario, the external vector is smaller for bigger cities, which implies that the slope of the fit line decreases over time. That is apparently the case for the water networks of Brazilian cities. One possible explanation is that the system is still out of equilibrium. That is, water networks in many cities in Brazil are not sufficiently developed yet, and might converge into equilibrium (when the magnitude of the external vector of all cities will be around an average value) given enough time.

It is interesting to see these facts through the lens of Pumain et al.'s theory [14], which says that a system of cities is formed through a hierarchical diffusion of innovations: innovation processes start in bigger cities and then diffuse to smaller ones. They would lead to different time dynamics for bigger and smaller cities concerning certain urban metrics, such as water network length. Interpreting our concept of the external vector through Pumain et al.'s theory, one could argue that some technologies required for water networks were initially deployed in bigger cities and then progressively diffused throughout the urban system. The relatively small

external vector for bigger cities would be saying that such technologies are still in the diffusion process stage. In any case, in order to have a better understanding of that process, further research is necessary, which can be achieved by following the evolution of more urban variables.

## Conclusions

Urban scaling has been seen as a crucial phenomenon to understand cities. For larger population sizes, critical variables like socioeconomic output would grow at more than proportional rates, while others like urban infrastructure would grow at less than proportional rates. The mechanism to explain this was linked to properties of networks at the heart of urban density and diversity, triggering increasing returns to scale in the economy and scale economies in infrastructure. This scaling pattern has been observed across large groups of different cities in particular periods ('transversal' or 'cross-sectional' scaling). However, would an individual city growing in time ('longitudinal' or 'temporal' scaling) follow the same scaling pattern? This paper searched for an answer to this question.

Transversal and longitudinal scaling patterns in cities are intuitively expected to converge. That seems to make sense, since urban scaling in both longitudinal and transversal contexts are intrinsically related phenomena tied by their statistical nature. But do they actually mirror each other or can be reduced to one another? Do they behave similarly even for different urban metrics? The present work assessed the extent of their compatibility, exploring the context of a developing economy. In short, we verified under which conditions transversal ($\beta_T$) and longitudinal ($\beta_i$) scaling exponents are similar and where we could expect discrepancies.

In order to analyze the conditions of compatibility between longitudinal and transversal scaling exponents, we introduced a theoretical formulation based on vector fields, first to separate *internal* from *external* factors active in the urban metrics, and then to yield a quantitative relationship between $\beta_T$ and $\beta_i$. This theoretical framework was then confronted with empirical evidence of the time evolution of GDP and water network length of Brazilian cities. We analyzed scaling dynamics of 5507 Brazilian cities and conurbations, aggregated as contiguous urban agglomerations from the totality of 5570 municipalities.

As a result, we found signs of a relationship between the longitudinal and transversal exponents, provided that certain conditions are met. The theoretical approach suggests that these exponents will be numerically equal only in urban systems under the following conditions: i) abstracting external factors in city dynamics; ii) stability of the transversal exponent throughout the system; and iii) cities with a significant growth rate.

The first condition can be understood as an ideal situation since in practice it would be impossible to find a city completely isolated from the rest of the system. The equation obtained allows us to conclude that in practice the longitudinal exponent for cities with sufficient growth will always be greater than the transversal exponent, a fact that is observed empirically. The second condition can be understood as a consequence of a mature or a stable system concerning a specific urban metric. The third condition establishes specific terms for the longitudinal scaling analysis to attain empirical sense. Therefore, in longitudinal analysis, scaling laws will only be properly manifested in cities with a sufficiently large population, and they are not suitable to be observed in small cities. This is a direct consequence of the fact that urban scaling laws, in both longitudinal and transversal contexts, are phenomena of an intrinsically statistical nature.

Our results suggest that longitudinal (temporal) scaling exponents are city-specific. However, they are distributed around an average value that approaches the transversal scaling exponent when the conditions above are met—but not for every urban metric. For example, in the

case of GDP, which has a relatively constant transversal scaling exponent in the analyzed period, the removal (decomposition) of external factors leads us to a distribution of longitudinal exponents around the expected value of the transversal exponent. On the other hand, in the case of the water network length, which may have a transversal scaling exponent out of balance in a developing economy, the decomposition of external factors leads us to a non-coherent situation between longitudinal and transverse scaling exponents. Adding more nuance to the scaling dynamics, the relationship between $\beta_i$ and $\beta_T$ for both GDP and water network length gets closer only when we consider cities with significantly high growth rates. This happens because cities with low population growth present unstable or divergent longitudinal scaling exponent values, adding noise or complexity to the relationship.

Finally, our mathematical formulation of vector fields, in line with Puiman et al. [14], provided a hypothesis about the temporal dynamics of water network length in a developing economy like Brazil. Our analysis showed that the transversal scaling exponent for this metrics has decreased in time, and that the vector associated with external factors is larger (in average) in smaller cities, suggesting that the diffusion of technologies involved in the construction of water networks have not fully reached small cities yet, being still underway. Based on Pumain et al.'s theory, we can argue that the system will converge ($\beta_T$ became constant) when the external vector for both smaller and greater cities are similar (in average), and the system comes into equilibrium, completing the diffusion process.

Theory and empirical evidence suggest that transversal and longitudinal scaling are intrinsically related, as the temporal evolution of individual cities leads to general states of the urban system. However, divergences between temporal and cross-sectional scaling exponents suggest that there is more complexity to the systemic behavior of cities than a single general scaling law could account for (cf. [47]). The longitudinal analysis allows us to see differences in the behaviour of cities within the overall pattern, particularly related to size: which urban population sizes are likely to obey or add instability to the scaling law. It also allows us to consider differences in scaling behavior related to the nature of urban variables and how they materialize in time. Finally, it makes room for the role of external factors such as public policies, state of development and macroeconomic behavior—important contextual factors that add complexity to the effects of scaling. Instead of asserting a hallmark for an emerging field, these findings open up new possibilities in the research of scaling effects, including further exploration of urban variables and regional contexts.

## SM1—Supplementary Material 1

One can find attached the spreadsheet generated by data mining of the website of the *Brazilian Institute of Geography and Statistics (IBGE) and the water-sewage-waste companies national survey (SNIS). In this spreadsheet, one can find population size, GDP (from the year 1999 to 2014), and water network length (from the year 1995 to 2015), for all 5570 Brazilian Municipalities. This spreadsheet has all the necessary data to replicate the results of our study in its entirety.*

## SM2—Supplementary Material 2

Video presenting the time evolution of the GDP for Brazilian Southwestern cities. The points that represent cities are in an apparently chaotic movement around the slop of the red line, obtained by the best fit of the power-law function, and maintaining itself practically constant. The video is available at: https://www.youtube.com/watch?v=2sP-J7fdN_c.

## Supporting information

**S1 Data.**
(ODS)

**S1 Video.**
(MKV)

## Acknowledgments

We are deeply saddened to share with the urban science community the unexpected passing of our coauthor and dearest friend Joao Meirelles. Joao was a young and kind person, and a most promising talent. We will miss you, Joao.

We would like to acknowledge all colleagues from the Mathematical Department of City, University of London, where most of the analytical work for this article was done. FLR acknowledges members from CASA-UCL, especially the stimulating discussions with Elsa Arcaute during his sabbatical year in 2017.

An earlier version of this work was published in arXiv on 4 Oct 2019.

## Author Contributions

**Conceptualization:** Fabiano L. Ribeiro, Joao Meirelles.

**Data curation:** Fabiano L. Ribeiro, Joao Meirelles.

**Formal analysis:** Fabiano L. Ribeiro, Joao Meirelles.

**Funding acquisition:** Fabiano L. Ribeiro.

**Investigation:** Fabiano L. Ribeiro, Joao Meirelles, Vinicius M. Netto, Camilo Rodrigues Neto, Andrea Baronchelli.

**Methodology:** Fabiano L. Ribeiro, Joao Meirelles.

**Project administration:** Fabiano L. Ribeiro, Joao Meirelles.

**Resources:** Fabiano L. Ribeiro, Joao Meirelles.

**Software:** Fabiano L. Ribeiro, Joao Meirelles.

**Supervision:** Fabiano L. Ribeiro, Andrea Baronchelli.

**Validation:** Fabiano L. Ribeiro, Joao Meirelles, Camilo Rodrigues Neto, Andrea Baronchelli.

**Visualization:** Fabiano L. Ribeiro, Joao Meirelles.

**Writing – original draft:** Fabiano L. Ribeiro.

**Writing – review & editing:** Fabiano L. Ribeiro, Joao Meirelles, Vinicius M. Netto, Camilo Rodrigues Neto, Andrea Baronchelli.

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
