## [Decision Letter · Decision Letter 0]

9 Jan 2020

PONE-D-19-29567

On the relation between transversal and longitudinal scaling in cities

PLOS ONE

Dear Mr. Meirelles,

Thank you for submitting your manuscript to PLOS ONE. After careful consideration, we feel that it has merit but does not fully meet PLOS ONE’s publication criteria as it currently stands. Therefore, we invite you to submit a revised version of the manuscript that addresses the points raised during the review process.

We would appreciate receiving your revised manuscript by Feb 23 2020 11:59PM. To enhance the reproducibility of your results, we recommend that if applicable you deposit your laboratory protocols in protocols.io, where a protocol can be assigned its own identifier (DOI) such that it can be cited independently in the future. For instructions see: http://journals.plos.org/plosone/s/submission-guidelines#loc-laboratory-protocols

We look forward to receiving your revised manuscript.

Kind regards,

Andrea Antonio Guido Caragliu

Academic Editor

PLOS ONE

Additional Editor Comments:

Decision letter PONE-D-19-29567

On the relation between transversal and longitudinal scaling in cities

Dear Prof. Meirelles, dear Joao:

Thank you for considering Plos ONE as a potential outlet for your research.

I have now received two referee reports on your paper, and I am happy to report that they agree the paper has several merits that bode well for its potential impact in the fields of econophysics and urban economics.

I also read your paper carefully and I would advise the following three relatively minor revisions, based on the attached reviewers’ comments and my own reading of your work.

• Reviewers agree that it is not fully clear whether the definition of the urban areas used in your analyses is a purely administrative or functional one, or a mix of the two. This is an important issue to be explained upfront for ensuring full replicability of your findings.

• Reviewer 1 points at a possible gap in your model predictions, in that your results would hold only when cities actually grow. Would you be able to find the same results were shrinking cities also taken into account? In other words: do these scaling laws apply to both (negative and positive) sides of the urban growth distribution?

• From a theoretical perspective, while your use of scaling laws derived from physics and in general hard sciences has been often found to fit well statistical distribution of city size and GDP growth, it would also be beneficial to better and more clearly position the paper within a Urban Economics perspective. The work of Henderson, Gabaix, and the theoretical models à la Fujita, Krugman, and Mori offer useful bridges in this respect.

I am looking forward to receiving your revised manuscript at your earliest convenience, and I thank you once again for thinking of Plos ONE as a suitable output for your work.

Kind regards,

Andrea Caragliu

Journal Requirements:

'This work was supported by Swiss Mobiliar (Joao Meirelles), École polytechnique fédérale de Lausanne (Joao Meirelles), Conselho Nacional de Desenvolvimento Científico e Tecnológico (Fabiano Lemes Ribeiro, process number: 405921/2016-0),  Coordenação de Aperfeiçoamento de Pessoal de Nível Superior (Fabiano Lemes Ribeiro, process number: 88881.119533/2016-01). The funders had no role in study design, data collection and analysis, decision to publish, or preparation of the manuscript.'

We note that you received funding from a commercial source: Swiss Mobiliar

Reviewers' comments:

Reviewer's Responses to Questions

**Comments to the Author**

1. Is the manuscript technically sound, and do the data support the conclusions?

Reviewer #1: Yes

Reviewer #2: Yes

2. Has the statistical analysis been performed appropriately and rigorously? 

Reviewer #1: Yes

Reviewer #2: Yes

3. Have the authors made all data underlying the findings in their manuscript fully available?

Reviewer #1: No

Reviewer #2: Yes

4. Is the manuscript presented in an intelligible fashion and written in standard English?

Reviewer #1: Yes

Reviewer #2: Yes

5. Review Comments to the Author

Reviewer #1: PLOS One Review for Article

On the relation between transversal and longitudinal scaling

in cities

The paper looks at scaling behavior for 5507 municipalities - individual municipalities within cities are not whole urban units, and therefore the geographic definitions should be argued for and justified.

In author summary: the authors write: "while others like urban

infrastructure would grow at a slower rate". this should be rewritten as "while others like urban infrastructure would grow but at less than proportional rates" (making the sublinear pattern clear). Similarly, "socioeconomic output would grow at a higher pace" should be something like "socioeconomic output would grow at a more than proportional rate".

Is a municipality the same as a neighbourhood within a city, or an entire city? Explain upfront.

The authors say "The 28

universality proposition has been challenged [26–28], but most evidence seem to confirm 29

the generality, while exceptions are normally explained by local 30

particularities [10, 18, 29, 30]." This is too strong and early to claim. For example, several research papers old and new show in fact that the population variable alone is insigniifcant in regressions when industrial or occupational organizations are considered, or statistical models other than the scaling model equally well explain the data. This statement thus needs to be softened - saying that this is as yet an open question, both domain wise and methodologically and theoretically:

Scaling and Hierarchy in Urban Economies, https://arxiv.org/abs/1102.4101

Evidence for localization and urbanization economies in urban scaling, https://arxiv.org/abs/1910.07166

Leitao, J. C., Miotto, J. M., Gerlach, M. and Altmann, E. G. 2016. Is this

scaling non-linear? Royal Society Open Science 3: 150649.

Again, this is too naive: the total amount of social interactions between its citizens would 33 guide, to a great extent, the city towards the observed scaling behavior: economic and social organizations could be very different given the same group sizes or interaction densities. This claim seems to suggest that a 100 school teachers interacting is the same as a 100 finance and bankers interacting, and it simply does not matter what is being interacted upon!

This 34

proposition is unprecedented in urban science and the identification and validation of 35

such universal dynamics could help urban policymakers to identify opportunities and 36

improve the life quality of dwellers.

The above is too vague and no claims on policy should be made without a deep and direct connection to how such (still dubious, open or unconfirmed) findings can actually help policy.

Overall, there is a crucial question that is missing here, and at least a qualitative discussion should be added to address this issue: the scaling equations as they currently stand, show only growth - which is fine as an approximation as far as transversal systems of cities are concerned. However, when it comes to individual cities, they can also "die" - in the sense that there is a process of growth, but there is also a process of decline and death. In such a case, do the authors hypothesise that declining populations and their corresponding urban variables would fit on the same law? Detroit? Etc.

Materials and Methods

Repeat but very important issue: for following point: The data presented here refer to 5507 Brazilian municipalities, with contiguous dense 79

surrounding areas aggregated in single spatial units from the totality of 5570 Brazilian 80

administrative divisions...please provide justification that these can be considered as individual cities (why? - daily commute lengths are contained / population numbers / densities...what is the basis for defining each as an individual city, and how is this a reliable basis). The results in the entire paper rest on this crucial point.

Figure 1: beta as 1.04 by the maximum likelihood method, which maximum likelihood method? Provide reference. Also, 1.04 is only mildly superlinear, so strong claims on super linearity will not hold.

Results:

The authors should present some hypotheses and discussions on why they think that GDP (which is an economic variable) and the water supply network, which is a physical variable, should show differing relationships when compared with the transversal.

They make the beginnings of this in considering city growth rates - but the main question is why does an economic variable behave different to a physical infrastructure variable?

Theoretical section:

The formal part presented in the theoretical section is clear and nicely laid out. However, what I was not able to understand is - there are also old and established economic theories that discuss the growth of cities (for example, the entire body of work by Henderson, Gabaix, Krugman, Thisse, Fujita, Ogawa....) - how does the current naive physics of vector fields (which is not very physically motivated, but is a very "data" based perspective) relate to the previous work done in detail on these questions? At present, the work feels overall like a naive exploration of data.

Reviewer #2: This paper describe the relation between what the authors called transversal and longitudinal scaling in cities, this is, the relationship of urban metrics with population for cross-sectional data (same time) and temporal data (same city evolving with time). This is a hot topic and has recently picked the attention of people working in the "science of cities" and I think this paper should be eventually published in some form, as it offers more insights into this problem. I think, however, some of the recent debates on this topic are missing in this version of the manuscript.

My only recommendations are the following:

1) The authors include a discussion about the following paper:

- The Interpretation of Urban Scaling Analysis in Time (https://papers.ssrn.com/sol3/papers.cfm?abstract_id=3459540);

2) The sort of similar study (though with a different perspective) also looked at the evolution of urban scaling for several urban indicators for Brazilian data.

- Scale-Adjusted Metrics for Predicting the Evolution of Urban Indicators and Quantifying the Performance of Cities (https://journals.plos.org/plosone/article?id=10.1371/journal.pone.0134862)

The authors could include a comment about this paper and explain the possible relation of its findings with your results.

There are some typos that should be fixed. For instance, page 7, "quantity Whose" or page 9, "the external Vector".

Apart from that, I am happy with the current version of the manuscript and a would recommend it for publication after fixing the above points.

6. PLOS authors have the option to publish the peer review history of their article (what does this mean?). If published, this will include your full peer review and any attached files.

Reviewer #1: No

Reviewer #2: No

---

## [Author Response · Author response to Decision Letter 0]

6 Apr 2020

We thank you warmly for the careful consideration of our manuscript. We have

addressed all of the Reviewers’ concerns as detailed in the pdf file attached.

---

## [Editor Report · Decision Letter 1]

28 Apr 2020

On the relation between transversal and longitudinal scaling in cities

PONE-D-19-29567R1

Dear Dr. Ribeiro,

We are pleased to inform you that your manuscript has been judged scientifically suitable for publication and will be formally accepted for publication once it complies with all outstanding technical requirements.

With kind regards,

Andrea Antonio Guido Caragliu

Academic Editor

PLOS ONE

Additional Editor Comments (optional):

Dear Dr. Ribeiro, dear Fabiano:

thank you for the revised version of your paper. I read it carefully and I am happy to report that in my view all comments raised in the first version you submitted have been successfuly dealt with. Consequently, I would be inclined to accept the paper as it is without any further rounds of revisions. Congratulations and thanks again for thinking of Plos ONE as a possible outlet for your research.

Kind regards,

Andrea Caragliu
---

## [Editor Report · Acceptance letter]

5 May 2020

PONE-D-19-29567R1 

On the relation between transversal and longitudinal scaling in cities 

Dear Dr. Ribeiro:

I am pleased to inform you that your manuscript has been deemed suitable for publication in PLOS ONE. Congratulations! Your manuscript is now with our production department. 

With kind regards,

on behalf of

Professor Andrea Antonio Guido Caragliu 

Academic Editor

PLOS ONE